# Nonlinear co-generation of graphene plasmons for optoelectronic logic operations

Yiwei Li[1,9], Ning An[1,9], Zheyi Lu[2,9], Yuchen Wang[1], Bing Chang[1], Teng Tan [1,3], Xuhan Guo[4], Xizhen Xu[5], Jun He[5], Handing Xia[6], Zhaohui Wu[6], Yikai Su [4], Yuan Liu [2✉], Yunjiang Rao [1,3✉], Giancarlo Soavi [7,8✉] & Baicheng Yao [1✉]

Surface plasmons in graphene provide a compelling strategy for advanced photonic technologies thanks to their tight confinement, fast response and tunability. Recent advances in the field of all-optical generation of graphene's plasmons in planar waveguides offer a promising method for high-speed signal processing in nanoscale integrated optoelectronic devices. Here, we use two counter propagating frequency combs with temporally synchronized pulses to demonstrate deterministic all-optical generation and electrical control of multiple plasmon polaritons, excited via difference frequency generation (DFG). Electrical tuning of a hybrid graphene-fibre device offers a precise control over the DFG phase-matching, leading to tunable responses of the graphene's plasmons at different frequencies across a broadband (0 ~ 50 THz) and provides a powerful tool for high-speed logic operations. Our results offer insights for plasmonics on hybrid photonic devices based on layered materials and pave the way to high-speed integrated optoelectronic computing circuits.

[1] Key Laboratory of Optical Fibre Sensing and Communications (Education Ministry of China), University of Electronic Science and Technology of China, Chengdu, China. [2] Key Laboratory for Micro-Nano Optoelectronic Devices (Education Ministry of China), School of Physics and Electronics, Hunan University, Changsha, China. [3] Research Centre for Optical Fibre Sensing, Zhejiang Laboratory, Hangzhou, China. [4] State Key Laboratory of Advanced Optical Communication Systems and Networks, Shanghai Jiao Tong University, Shanghai, China. [5] Guangdong and Hong Kong Joint Research Center for Optical Fiber Sensors, Shenzhen University, Shenzhen, China. [6] Research Center of Laser Fusion, China Academic of Engineering Physics, 621900 Mianyang, China. [7] Institute of Solid State Physics, Friedrich Schiller University Jena, Jena, Germany. [8] Abbe Center of Photonics, Friedrich Schiller University Jena, Jena, Germany. [9] These authors contributed equally: Yiwei Li, Ning An, Zheyi Lu. ✉email: yuanliuhnu@hnu.edu.cn; yjrao@uestc.edu.cn; giancarlo.soavi@uni-jena.de; yaobaicheng@uestc.edu.cn

Nano-photonic technologies hold great promises for high-speed signal processing, as they circumvent light-electron conversion and offer the ideal solution to the problems of heat generation and limited bandwidth that are typical of electronic circuits[1,2]. In recent years, intense research in the field of optical computing has led to novel and advanced solutions for high-speed modulation and demodulation[3–5], metamaterial analog processing[6,7], optoelectronic convolution[8,9], machine learning[10–13] and quantum computing[14]. Since logic gates are the most fundamental building-block of classical and quantum computing[15], their development is nowadays of paramount importance. While classical electrical computing is based on transistors, a gold standard for optical logic gates is still elusive. For instance, by exploiting the linear interference in micro-rings and micro-couplers, optical logic operations have been successfully obtained on-chip[16–20]. However, the diffraction limit[21] hinders further miniaturization of such devices and creates a bottleneck for industrial expansion.

On the other hand, due to their high resonant frequencies (≈tens of THz) and unique field confinement (≈hundreds of nm)[22,23], graphene plasmons are promising candidates for the realization of miniaturized photonic integrated signal-processing elements beyond the diffraction limit of light[24,25]. In addition, in contrast to noble metals, graphene's plasmons display flexible electrical tunability[26–28], which in turn offers a powerful knob for dynamic information processing. Thanks to the strong nonlinear response of graphene[29–32], such electrically tunable plasmons can be generated all-optically and subsequently manipulated and detected directly on-chip[33,34], providing a viable strategy for planar integration in high-density optoelectronics. The main downside of the conventional methods that are used for the nonlinear generation of plasmons in graphene is the necessity of a careful laser scanning with a tunable range up to 10 THz, which is typically slow and has limited bandwidth and thus prevents the realization of multiple plasmonic generations for ultrafast logic operations. For instance, in ref. [33], only one graphene plasmon mode with a single terahertz frequency was generated.

Here, we exploit difference-frequency-generation (DFG) to demonstrate multiple plasmons co-generation in a hybrid graphene photonic device. In this scheme, two synchronized and stabilized mode-locked lasers (i.e. laser frequency combs)[35,36] are counter-propagating inside a hybrid graphene/D-shaped fibre and are used as pump and probe pulses for the DFG in a broad frequency range up to 50 THz. In particular, we simultaneously detect multiple plasmon peaks thanks to the broadband frequency-momentum conservation. Finally, by electrically tuning the graphene's Fermi level, we show individual modulation of each of the co-generated plasmons, thanks to the distinct phonon-plasmon interactions in different branches. This provides a powerful method to realize different logic operations (AND, OR, NOR) on a single integrated photonic device.

## Results

### Synchronized combs based nonlinear generation of plasmons.
Figure 1a shows a sketch of the graphene/D-shaped fibre (GDF) used in this work. A section of a silica single-mode fibre was side-polished, and a monolayer graphene was deposited on top of the planar surface (see Methods for details). Two gold electrodes (channel width ≈ 200 μm) are used for the electrical tuning of the graphene's Fermi level[37]. The fibre core is 6 μm in diameter and light-graphene interaction occurs via evanescent waves[38]; the total linear losses of the graphene-fibre heterostructure are <0.5 dB at 1560 nm when the $|E_F|$ of graphene is >0.4 eV. Clearly, by tuning the graphene Fermi level, the transmission loss of the device will change. Further details on the nanofabrication and

characterization of the GDF are provided in Supplementary Note 2. In the DFG process used for plasmon generation, graphene acts both as the second-order nonlinear medium and as the nanoscale plasmon waveguide. Driven by the out-of-plane second-order nonlinear susceptibility $\chi^{(2)}$ of the GDF, the high-frequency pump photon ($f_{pump}$) splits into a lower frequency probe photon ($f_{probe}$, counter-propagating) and a plasmon ($f_{sp}$, co-propagating). The DFG requires both energy conservation $f_{sp} + f_{probe} = f_{pump}$ and momentum conservation $\mathbf{k_{pump}} = \mathbf{k_{sp}} + \mathbf{k_{probe}}$, where $\hbar\mathbf{k_{sp}}$, $\hbar\mathbf{k_{probe}}$ and $\hbar\mathbf{k_{pump}}$ are the momenta of the plasmon, probe and pump, respectively. Considering the scalar dispersion relation $k = 2\pi f n/c$, the momentum conservation can be re-written as $f_{sp} n_{sp} = f_{pump} n_{pump} + f_{probe} n_{probe}$. In order to generate plasmons in graphene via the DFG process, all the optical modes must have transverse magnetic (TM) polarization that probes the non-centrosymmetric out-of-plane direction of the hybrid device, while graphene is centrosymmetric for in-plane excitation[39,40]. As an example, Fig. 1b simulates the side-view electric field distributions (TM polarization, normalized intensity) of the pump and the plasmon modes when $f_{pump} = 192$ THz and $f_{sp} = 10$ THz.

Figure 1c shows a top-view optical image of the GDF. The fibre core is denoted by a white dashed line and graphene is deposited on the fibre and connected via the source and drain electrodes (Au). Unlike typical back-gate field-effect-transistors[41], this on-fibre Au-graphene-Au transistor is driven by current rather than gate voltage[42]. Within the Drude's model, the surface-plasmon frequency $f_{sp}$ is defined by the graphene's dispersion relation which, in turn, depends on the Fermi level: a higher carrier density induces a higher $f_{sp}$. In our GDF architecture, by changing the source-drain voltage ($V_D$) in the range 0 to 1 V, we can tune the effective graphene Fermi level $|E_F|$ from ≈0 to 0.4 eV. Figure 1d plots the measured device resistance and the calculated $|E_F|$ as a function of $V_D$. When $V_D = 0$ V graphene is intrinsically positively charged (p-doped) with $|E_F| ≈ 0.2$ eV. By increasing $V_D$ we drive a current in the graphene channel and shift $E_F$ towards the Dirac point and the upper Dirac cone (n-doping). The graphene's resistance reaches the maximum (Dirac point) when $V_D = 0.16$ V. We notice that it is experimentally impossible to achieve $E_F = 0$ eV due to the presence of puddles and inhomogeneities[43]. However, for simplicity we define $E_F = 0$ eV as the point where we measure the minimum conductivity in the GDF. Such behaviour is typical of a graphene bipolar junction[38,42], as discussed in detail in Supplementary Note 2. To confirm the $V_D$ dependence of $E_F$, we have also characterized the doping via in-situ Raman spectroscopy (see Supplementary Note 3). Besides, due to $V_D$ is in ±1 V, temperature alteration induced by the Joule heating effect[44] is negligible, verified by our thermal imaging measurements (Supplementary Fig. 11).

Figure 1e plots the two frequency combs that are used for the DFG experiments. Comb 1 (blue curve, pump) is a stabilized Er mode-locked laser with a tunable central wavelength ≈ 1560 nm, 3 dB bandwidth ≈ 7.5 THz and maximum average power of ≈30 mW. Meanwhile, Comb 2 (red curve, probe) is spectrally flat in the wavelength region ≈ 1500–2100 nm and it is obtained by supercontinuum generation starting from Comb 1 (see Methods for details). The two combs are locked with the same repetition rate of ≈38 MHz and their temporal overlap can be controlled experimentally by a delay line. In the DFG experiments, the two combs are launched into the GDF from opposite directions (the experimental setup is shown in Supplementary Note 3). We further note that in order to achieve co-generation of multiple plasmons and detect them accurately, the pump should be spectrally sharp while the probe must be broadband and spectrally flat. The following scenario can now occur: for those frequencies that satisfy the phase-matching condition, the simultaneous presence (time-overlap) of the pump (Comb 1)

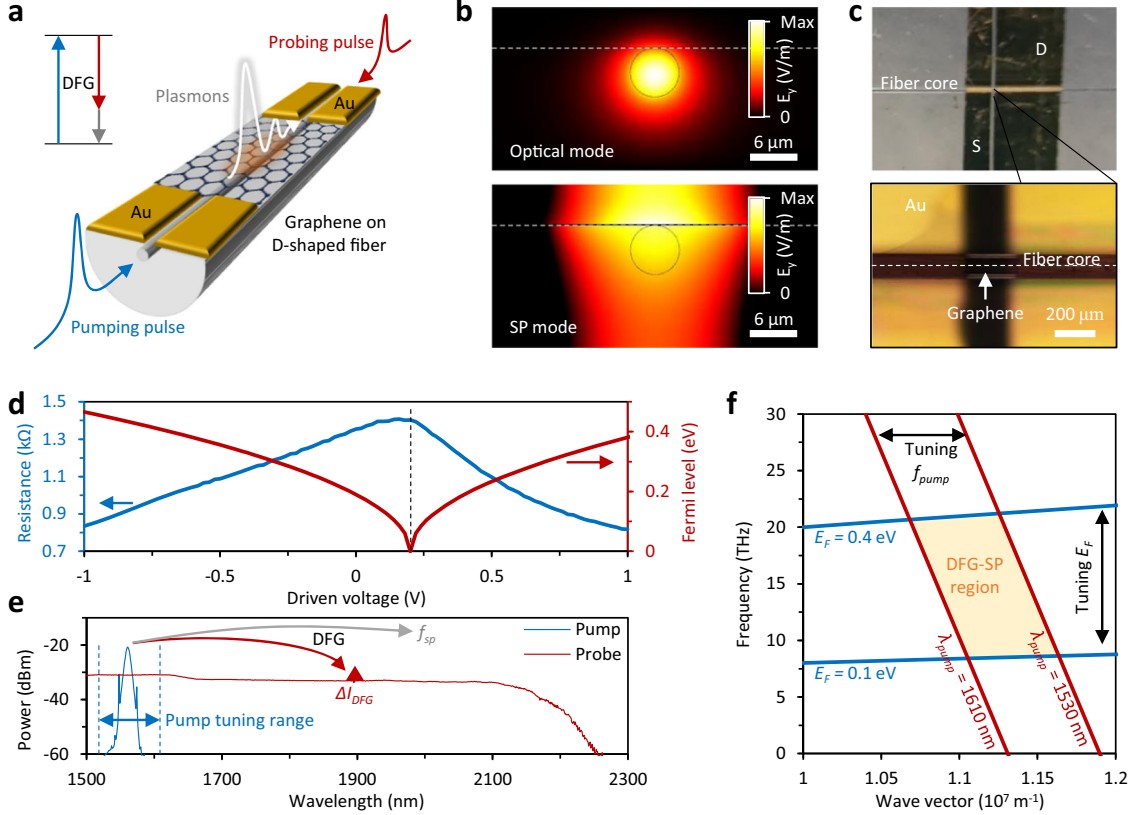

**Fig. 1 Generation and control of graphene's plasmons in a graphene heterogeneous fibre device. a** Schematic of the graphene optoelectronic device deposited on a D-shaped fibre. DFG via the surface $\chi^{(2)}$ nonlinearity is excited by two counter-propagating synchronized pulses. DFG difference-frequency generation. Inset: energy conservation for the DFG process. **b** Simulated electrical field distributions of an optical mode propagating in the fibre core and a plasmonic mode propagating along the graphene-fibre interface (transverse magnetic polarization, $E_y$). SP surface plasmon. Scale bar: 6 μm. The horizontal dashed line shows the fibre-air interface. Colour bar: normalized $E_y$. **c** Top-view microscope image of the graphene transistor on-fibre. The white line shows the fibre core, the monolayer graphene is connected by two Au electrodes, with contact channel length 200 μm. **d** Measured resistance and calculated Fermi level, the sampling rate is 20 mV. **e** Spectra of the pump comb (blue) and the probe comb (red). The red and grey arrows illustrate the DFG process. $f_{sp}$ the frequency of surface plasmon, $\Delta I_{DFG}$ the DFG enhanced signal intensity. **f** Parametric space of the DFG plasmons. The blue curves show the electrically tunable dispersion of graphene, while the red curves show the optically tunable phase-matching condition. $f_{pump}$ the frequency of pump, $E_F$ the graphene's Fermi level, the shaded area demonstrates the possible region of the graphene nonlinear generation in our experiment.

and the probe (Comb 2) will lead to the generation of a plasmon and to the enhancement of the counter-propagating probe (Comb 2). Thus, the plasmon generation and the DFG process can be detected as an increase of the probe intensity at a specific phase-matched frequency ($\Delta I_{DFG}$), in analogy with the widely used process of optical parametric amplification[45].

For example, for a pump wavelength of 1560 nm (192.308 THz) and a nonlinearly excited plasmon at $f_{sp} = 10$ THz, we will observe a peak on the flat spectrum of *Comb 2* (probe enhancement) at $\approx1645.57$ nm (182.308 THz). To further prove that this peak arises from DFG, we modulated the pump at 500 kHz and observed the same modulation in the counter-propagating probe comb (see Supplementary Note 3). In contrast to conventional schemes that use continuous-wave tunable lasers for plasmon generation[33], our two-combs approach does not require time to scan the laser wavelengths and operates with $\approx 2$ orders of magnitude less average optical power. More importantly, due to the large bandwidth (50 THz) of the probe comb, this scheme enables us to find high-frequency plasmons beyond the limitation of any near-infrared tunable lasers.

Figure 1f shows the parametric space of the DFG process. To generate the THz graphene plasmons, the phase-matching condition for the counter-pumped DFG and the dispersion of the plasmonic modes must match. For free-standing graphene,

the plasmonic dispersion is defined by the Drude's model ($k_{sp}\propto f_{sp}^2$):[25] The blue curves in Fig. 1f are exemplary cases when $|E_F| = 0.1$ eV and 0.4 eV. On the other hand, the phase-matching condition for DFG can be re-written as $(c/2\pi)k_{sp} = -f_{sp}n_{probe} + f_{pump}(n_{pump} + n_{probe})$. When tuning the pump comb from lower to a higher frequency (e.g. from 1530 to 1610 nm as shown in Fig. 1f), the red curve moves from right to left. Since graphene plasmons are generated only at the intersections of the graphene dispersion curves and the DFG phase-matching lines, by tuning the graphene's Fermi level and the pump frequency, the plasmons' frequency will shift within the yellow region of Fig. 1f. See experimental verification in Supplementary Note 3.

**Electrically tunable multiple graphene plasmons.** Figure 2 shows the electrical tunability of our device. In the measurements, we have a central $f_{pump} = 192.3$ THz (1560 nm) while the probe pulse is broadband. Figure 2a–c shows the graphene's plasmon dispersion calculated using the random phase approximation (RPA)[23] at $|E_F| = 0.1$, 0.2 and 0.3 eV, respectively. Here, we considered two surface-optical phonon resonances[46,47] of the silica substrate (fibre), located at $f_1 = 24$ THz and $f_2 = 35$ THz (white solid lines). The phase-matching condition for the DFG is marked by the white dashed line and we used the refractive inside

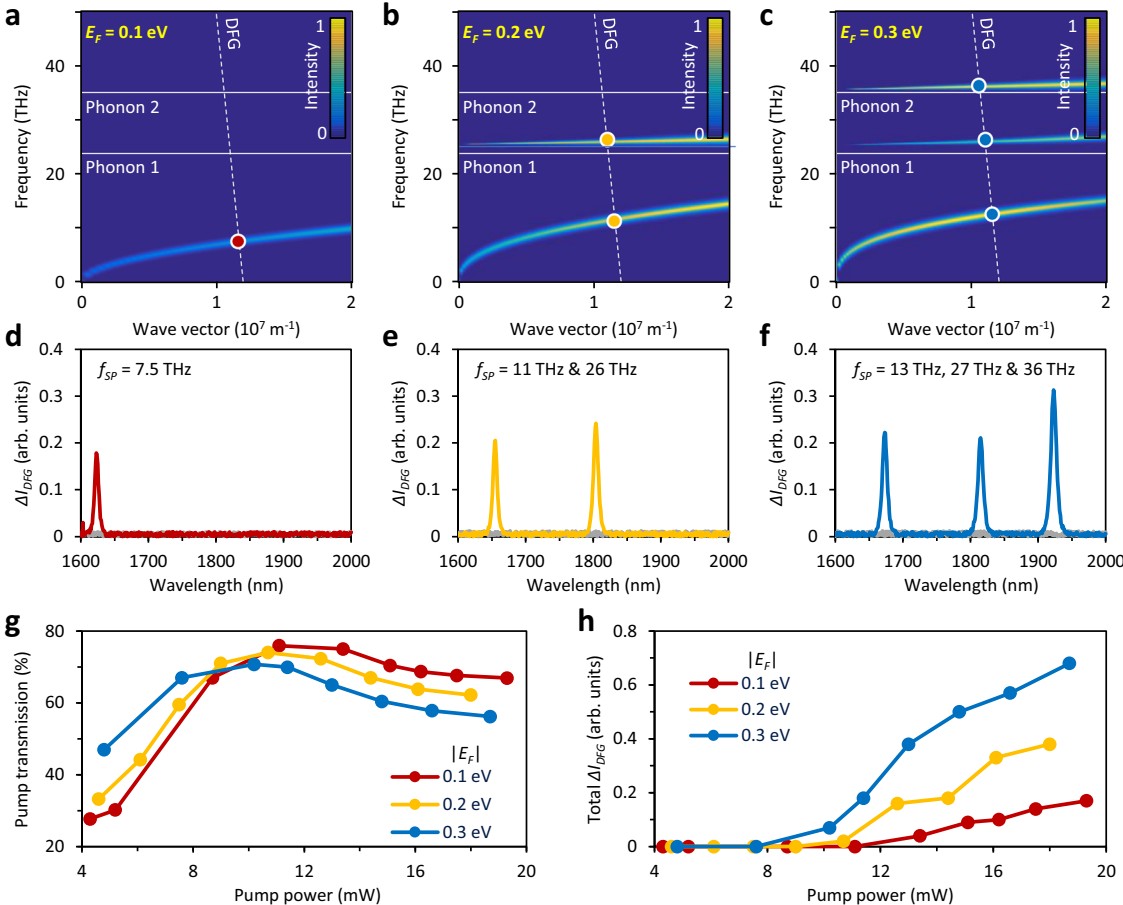

**Fig. 2 DFG signal and electrically tunable plasmons. a–c** Phase-matching condition (white dashed line) and graphene's plasmonic dispersion for $|E_F| = 0.1$, 0.2 and 0.3 eV. The red, yellow and blue dots mark the 'frequency-momentum' position of the nonlinearly generated plasmons. White solid lines: Phonon resonances of the silica substrate. Colour bar: normalized intensity. **d–f** Measured $\Delta I_{DFG}$ peaks. Red, yellow and blue curves plot the cases when $|E_F| = 0.1$, 0.2 and 0.3 eV, respectively. In these panels, the grey curves show the results measured in a D-shaped fibre without graphene for comparison. Colour bar of the maps shows normalized plasmonic intensity. **g** Pump transmission in the GDF device. The pump transmission first increases due to saturable absorption, and then decreases due to nonlinear DFG consumption. **h** The total $\Delta I_{DFG}$ increases linearly with the pump power.

the fibre's core $n_{pump} \approx n_{probe} = 1.45$. When $|E_F| = 0.1$ eV (Fig. 2a and d) the graphene's plasmon is far below the silicon phonon frequency, hence we can only see one DFG peak at 1623 nm ($f_{sp} = 7.5$ THz). When $|E_F| = 0.2$ eV (Fig. 2b, e) the graphene's plasmon interacts with the 24 THz phonon resonance, dividing the Drude curve into two branches. In this case, we observe two DFG peaks located at 1655 and 1804 nm ($f_{sp} = 11$ THz and 26 THz). Finally, when $|E_F| = 0.3$ eV (Fig. 2c, f) hybridization of the graphene's plasmon with the substrate phonons leads to three branches and we observe enhanced DFG peaks at 1672, 1813 and 1923 nm ($f_{sp} = 13$, 27 and 36 THz). In addition, based on the dispersion relation $n_{sp} = (f_{pump}n_{pump} + f_{probe}n_{probe})/(f_{pump} - f_{probe})$, we can estimate the effective refractive index of the plasmonic modes. For instance, when $f_{sp} \approx 7.5$, 27 THz and 36 THz we obtain $n_{sp} \approx 68$, 19 and 14, suggesting strong confinement of the plasmons. The simulated fibre mode indices are shown in Supplementary Note 1.

For a D-shaped fibre without graphene, we neither observe any plasmon generation nor DFG. This indicates that the process originates from graphene instead of from the gold contacts (see Supplementary Note 3). In order to further confirm the nature of the observed signal and distinguish it from other possible side effects (e.g. saturable absorption), in Fig. 2g, h we analyze the conversion efficiency of $\Delta I_{DFG}$. In particular, the generation of plasmons and the consequent intensity enhancement at $f_{probe}$ rely on the consumption

of the pump. We thus fix the probe power to 10 mW while increasing the pump power, and we measure both the pump transmission and the total $\Delta I_{DFG}$ at different values of $|E_F|$. For low pump power values (<8 mW), the curves at $|E_F| = 0.1$ eV and 0.2 eV are almost identical, while the curve at $|E_F| = 0.3$ eV shows higher transmission, likely due to reduced absorption in the thermally broadened Fermi Dirac distribution (considering the pump photon energy of ≈0.8 eV). Initially, the transmission curves for all $|E_F|$ values increase with pump power and reach a plateau at ≈12 mW due to saturable absorption[48]. Subsequently, for pump powers >12 mW, graphene is fully saturated and a further increase of the power leads to a lower transmission due to pump consumption via the DFG process. For the same value of the pump power (>12 mW) we observe that the $\Delta I_{DFG}$ raises from the noise floor and subsequently increases linearly (Fig. 2g, h), as expected for DFG considering $I_{sp} = \Delta I_{DFG}(f_{sp}/f_{probe}) = [\chi^{(2)}]^2 I_{pump} I_{probe}/L_{sp}^2$, where $L_{sp}$ is the $E_F$ dependent transmission loss of the plasmon (see ref. [33] and Supplementary Note 1). Specifically, when the phase-matching condition is satisfied, for $|E_F| = 0.1$, 0.2 and 0.3 eV, $L_{sp} \approx 1.67$, 1.18 and 1.04. For instance, in experiments at $|E_F| = 0.3$ eV, $\chi^{(2)}$ reaches $10^{-2}$ m/V, and $[\chi^{(2)}]^2/L_{sp}^2$ is on sub $10^{-4}$ W$^{-1}$ level, in agreement with theoretical values[30].

In Fig. 3 we show the ultrafast nature of our method by scanning the delay between the pump and probe combs while measuring the DFG enhanced signal ($\Delta I_{DFG}$), in analogy with an

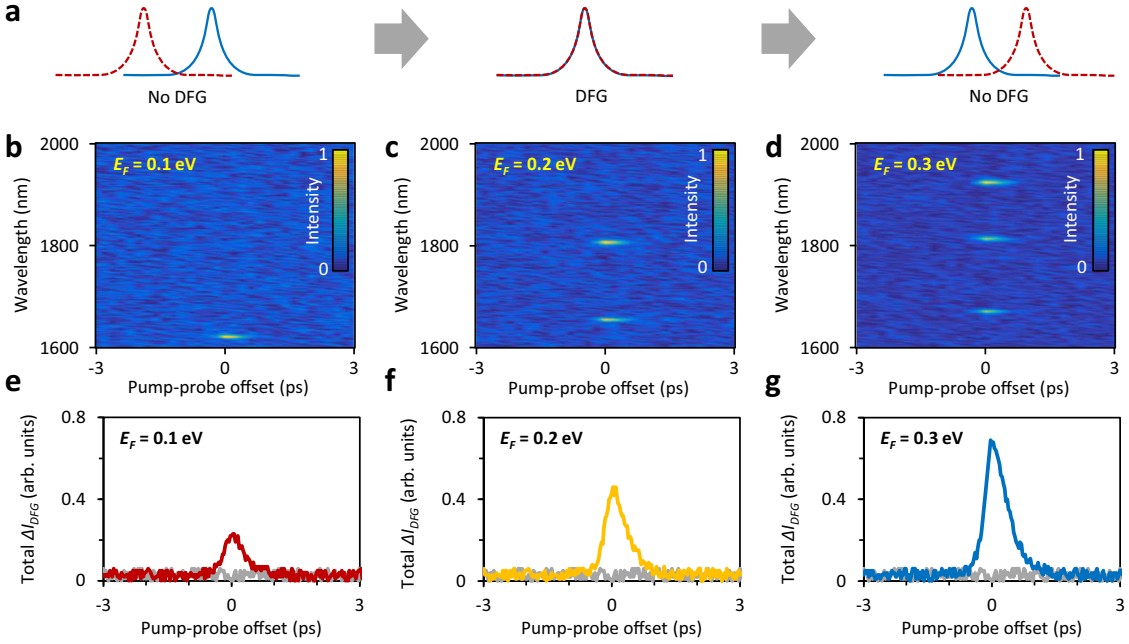

**Fig. 3 Time-correlation measurements of the plasmon generation via DFG. a** Sketch of the time-correlation experiment, DFG occurs only when the pump and the probe overlap in time. **b–d** The $\Delta I_{DFG}$ maps corresponding to different $f_{sp}$, when the graphene's Fermi level is 0.1, 0.2 and 0.3 eV. Colour bar: normalized intensity. **e–g** The total $\Delta I_{DFG}$ traces, when the graphene's Fermi level is 0.1, 0.2 and 0.3 eV. The grey curves show the reference signal (background) when $f_{pump} - f_{probe}$ is far from the plasmonic resonances.

auto-correlation measurement (Fig. 3a, see also the implementation in Supplementary Movie 1). Figure 3b–g shows the time-dependent DFG for $V_D = 0$ V ($|E_F| = 0.1$ eV), $V_D = 0.4$ V ($|E_F| = 0.2$ eV) and $V_D = 0.6$ V ($|E_F| = 0.3$ eV), respectively. At zero delay time we observe the largest nonlinear enhancement $\Delta I_{DFG}$ for all $E_F$ values. In this measurement, the central pump wavelength is 1560 nm and the time delay is tunable from −5 to +5 ps. The soliton pulse width for the pump and probe combs are ≈ 430 fs and ≈ 110 fs, respectively (Supplementary Note 3). The enhanced probe intensities are plotted in Fig. 3e–g: all the $\Delta I_{DFG}$ peaks have a time-width of ≈ 500 fs in sech$^2$ fitting, as expected from a parametric process given the pulse duration of the pump and probe pulses used in our experiments. Such measurements verify the fast response of the DFG in our GDF device, demonstrating a unique capability for high-speed optoelectronic operations.

**Electro-optic logic operations.** Finally, the co-existence of multiple electrically tunable plasmons allows us to perform logic operations, as schematically shown in Fig. 4a. Two parallel electrical signal generators ($V_A$ and $V_B$, the other contact is grounded) are used as input and each of them can provide either 0 V (OFF state, digital signal 0) or 0.5 V (ON state, digital signal 1). By combining the different states of $V_A$ and $V_B$ (i.e. both ON, only one ON or both OFF), the graphene's $E_F$ can be tuned to 0, 0.24 and 0.4 eV. The pump and probe beams are launched into the GDF in opposite directions and we detect the gate-tunable plasmon generation via the DFG as an increase in the probe intensity ($\Delta I_{DFG}$) at specific frequencies. For logic operations, three $\Delta I_{DFG}$ peaks (defined by the three plasmonic dispersion branches) with $f_{sp}$ in the 0–50 THz band are filtered using three bandpass filters (BPF) based on-fibre Bragg gratings (the spectral characterization of the filters is shown in Supplementary Note 3).

Figure 4b explains the filtering scheme more in detail in the retrieved '$E_F$-$f_{sp}$' map. When increasing the $|E_F|$ from 0.1 eV to 0.4 eV, the $f_{sp}$ of the three branches shifts in different ways. For

instance, the low-frequency branch shifts from 7.5 THz to 12.2 THz, while the middle and high-frequency branches experience almost no shift (<1.5 THz). On the other hand, as previously discussed (see Fig. 2), the middle frequency branch at 26–27 THz appears only for $|E_F| > 0.15$ eV, while the high-frequency branch at ≈ 37 THz appears for $|E_F| > 0.25$ eV due to plasmon-phonon interaction. By selecting the filtering frequency to 7.5, 27 and 37 THz with respect to the pump frequency, we can thus obtain the three different optical logic outputs. In particular, the $\Delta I_{DFG}$ from BPF3 is detectable only when both $V_A$ and $V_B$ are 'ON' (AND gate); the $\Delta I_{DFG}$ from BPF2 is detectable when either $V_A$ or $V_B$ is 'ON' (OR gate) while the $\Delta I_{DFG}$ from BPF1 is detectable only when both $V_A$ and $V_B$ are 'OFF' (NOR gate). Figure 4c shows a measured example of the optical logic operations. We design two square-wave modulated signal traces for both $V_A$ and $V_B$ (0.5 V RZ code, sampling rate 1 MHz, data stream 100 kbps). Accordingly, the different BPFs provide at the output the logic operation expected from the AND, OR and NOR gates. Specifically, the signal-to-noise ratios of the AND, OR and NOR gate outputs are higher than 89%, 82% and 77%, respectively, thus allowing precise discrimination of the logic operation. In future, the plasmonic co-generation and control scheme demonstrated here could be easily extended to on-chip integrated graphene optoelectronic devices suitable for large-scale fabrication, such as silicon nitride waveguides[33] and microrings[49]. Recent advances in the field of heterogeneously integrated microcombs on silicon[50] also provide a powerful tool to exploit pulsed dual-comb optical sources for the on-chip generation and control of the multiple graphene plasmons generated by DFG, further boosting the practicality of this scheme on versatile device platforms.

## Discussion
In conclusion, we used two synchronized counter-propagating laser combs to excite multiple electrically tunable plasmons via difference-frequency generation and parametric amplification in a

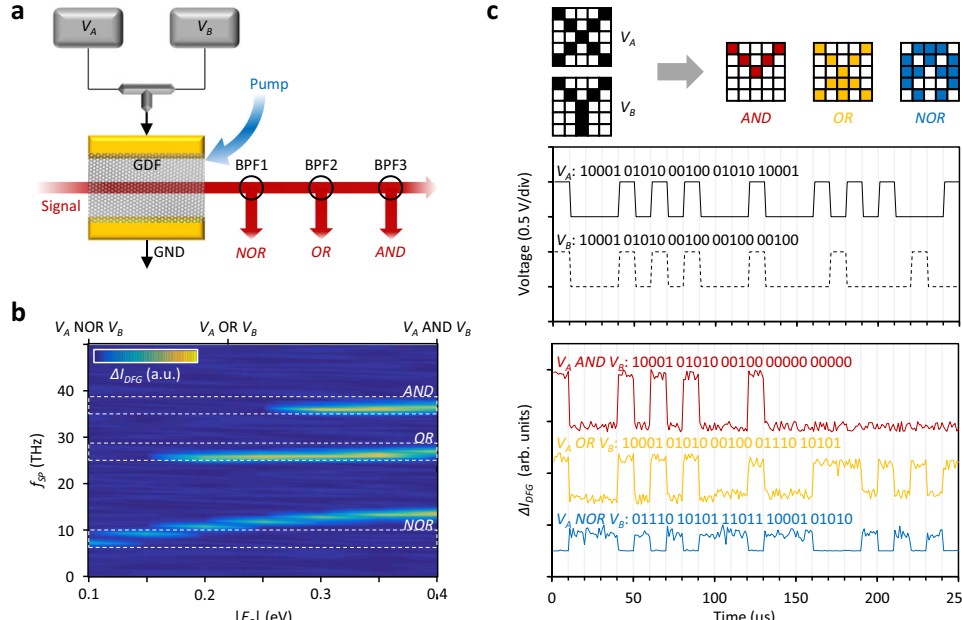

**Fig. 4 Electro-optic logic operations and gates. a** Schematic design of the setup to obtain the logic outputs NOR, OR and AND using three bandpass filters (BPFs). Here $V_A$ and $V_B$ are two independent digital signals used to tune the graphene $|E_F|$. **b** Results of the filtering in the $|E_F| - f_{sp}$ space for the frequencies 7.5, 27 and 36 THz selected by the three BPFs and corresponding to the three logic operation of NOR, OR, AND. Dashed boxes: filtering bands of the three BPFs. **c** Example of logic operations measured at the outputs of the three BPFs, here the black curves plot the amplitudes of $V_A$ and $V_B$, while red, yellow and blue curves show the output of the AND, OR, NOR gates.

graphene/D-shaped fibre device and demonstrate their potential for integrated logic operations. By electrically tuning the graphene's Fermi level, the co-generated plasmons in the frequency range 7.5–37 THz can be independently controlled and, after proper filtering, they can be used to realize three optoelectronic logic gates (AND, OR and NOR) in one single device. The optical generation and electrical control of multiple graphene's plasmons provide a platform for nanoscale integrated optoelectronic devices with a high potential impact in advanced applications such as ultrafast light-field manipulating, signal processing and optoelectronic computing.

## Methods

**Theoretical analysis.** The electrical field distribution inside the GDF was modelled using the finite element method. The graphene plasmonic dispersion and phonon coupling were calculated within the random phase approximation. The $\chi^{(2)}$ of graphene is estimated using the density matrix equations. More details are discussed in Supplementary Note 1.

**Fabrication of the graphene heterogeneous D-shaped fibre.** The D-shaped fibre samples were prepared by side-polishing a commercial single-mode silica fibre. The length of the D-shaped region is 2 mm and the insertion loss of the D-shaped fibre at 1560 nm is <1 dB. Single-layer graphene (SLG) was grown via CVD on Cu foil (99.8% pure) and then transferred onto the D-shaped fibre by using the polymethyl methacrylate (PMMA)-based wet transfer method. After bathing the PMMA/SLG/D-shaped fibre in acetone and water, the PMMA was removed. Then 30 nm/10 nm Au/Ti electrodes were deposited via the masked sputtering method. The effective area of the on-fibre graphene interacting with light is $10 \times 200\ \mu m$. Details are shown in Supplementary Note 2.

**Optical frequency comb sources and experimental setup.** A stabilized mode-locked fibre laser with a central wavelength 1560 nm, fixed repetition 38 MHz and pulse width $\approx 300$ fs is divided into two paths. The first one has 3 dB spectral bandwidth $\approx 6$ nm and it is used as the pump, while the second one is supercontinuum-broadened to cover the spectral range 1560–2100 nm and it acts as the probe. To ensure the highest accuracy during measurement, the frequency combs are stabilized with total relative intensity noise $< -120$ dB and phase noise $< -150$ dBc/Hz at 10 kHz. For the DFG plasmon detection we carefully implemented the following points: (1) both the pump and the probe are TM polarized, in order to maximize the graphene-light interaction and to probe its out-

of-plane direction; (2) a tunable free-space delay line is used to precisely control the counter-launched pulses, with motion accuracy <1 μm; (3) the maximum peak power of the pump reaches 10 kW, much higher than the graphene saturable absorption threshold. The different logic gates (AND, OR and NOR) are selected via three different bandpass filters based on-fibre Bragg gratings with a slightly tunable ($\approx 10$ nm) central wavelength for the optimization of the output intensity. Details of the experimental setup are shown in Supplementary Note 3.

## Data availability

The data that support the plots within this paper and other findings of this study are available from the corresponding author upon request. Source data are provided with this paper.

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

## Acknowledgements

We acknowledge support from the National Science Foundation of China (61975025, 51991340, U2130106), the Key project of Zhejiang Laboratory (2020KFY00562) and the National Key Research and Development Program (2021YFB2800602, 2021YFA1200503). This work was also supported by the European Union's Horizon 2020 research and innovation program under Grant Agreement GrapheneCore3 881603. G.S. acknowledges the German Research Foundation DFG (CRC 1375 NOA project B5) and the Daimler und Benz foundation for financial support.

## Author contributions

B.Y. led this research and Y.R. led the team. Yuan L. and B.Y. led the device design and fabrication. Yiwei L., N.A., Y.W., B.C. and T.T. performed the experiments. Z.L. and Yuan L. contributed to the graphene heterostructure fabrication and to the electrical measurements. B.Y., H.X. and Z.W. prepared the laser comb source. J.H. and Y.W. provided the D-shaped fibre samples and optimized the device. X.X. and J.H. contributed to the optical filtering. B.Y., X.G., Y.S. and G.S. performed the physical analysis. All authors processed and analyzed the results. B.Y., Yiwei L., Yuan L., G.S. and Y.R. prepared the manuscript.

## Competing interests

The authors declare no competing interests.
