## [Peer Review File · Nature Communications]

Nonlinear co-generation of graphene plasmons for optoelectronic logic operationsResponse to referees (NNANO-21122841)

Reviewer #1 (Comments for the Author):

>> The manuscript by Yiwei Li et al, reports on the use of differential frequency generation (DFG) to excite graphene plasmons in a photonic device showing potential for logic operations. The manuscript is very well written, with clear explanations of the physics and experiments performed, showing solid indications of the cogeneration of tunable graphene plasmons. I have no doubts about the results obtained.

We thank the Referee for the nice summary of our work and for his/her positive evaluation.

>> My main concern is the novelty of the work. In terms of physics, DFG of graphene plasmons has already been reported by some of the co-authors of this work.

We agree with the Referee that the basic principle of DFG of graphene plasmons has been recently reported [*Constant, T. et al. Nat. Phys. 12, 124, 2016; Yao, et al. Nat. Photonics 12, 22, 2018*]. However, we would like to stress that in this manuscript we experimentally demonstrate the **co-generation of multiple and tunable** graphene plasmons with a frequency comb based DFG process, which breaks the band-limit of the optical source, and to the best of our knowledge, demonstrates completely new results. In addition, the novelty of this work stems from the fact that we exploit this unique approach to realize a compact device for logic operations. We now further highlight the novelty of our work in the revised version of the manuscript.

We have highlighted this point in the revised version of the manuscript:

“The main downside of the conventional methods that are used for nonlinear generation of plasmons in graphene is the necessity of a careful laser scanning with tunable range up to 10 THz, which is typically slow and has limited bandwidth and thus prevents the realization of multiple plasmonic generation for ultrafast logic operations. For instance, in Ref. [33], only one graphene plasmon mode with single terahertz frequency was generated.”

And,

“In particular, we simultaneously detect multiple plasmon peaks thanks to the broadband frequency-momentum conservation. Finally, by electrically tuning the graphene’s Fermi level, we show individual modulation of each of the co-generated plasmons, thanks to the distinct phonon-plasmon interactions in different branches.”

>> On the other hand, the device presented in the paper cannot be considered "nano", since the sizes of all the elements (except for the thickness of graphene) are well in the microscale (although the introduction of the paper highlights the promises of nanophotonic technologies and the need to overcome the diffraction limit).

We thank the Referee for the comment. We would like to stress that the possibility to overcome the diffraction limit is related to the confinement of plasmons, not necessarily to the size of the device. On the other hand, the same approach demonstrated in this work can in be in future applied to other types of compact photonic devices, such as waveguides and microrings. In the revised version of the manuscript, we discuss this point with more details.

We have added the following paragraph to the revised version of the manuscript:

“In future, the plasmonic co-generation and control scheme demonstrated here could be easily extended to on-chip integrated graphene optoelectronic devices suitable for large-scale fabrication, such as silicon nitride waveguides [33] and microrings [49]. Recent advances in the field of heterogeneously integrated microcombs on silicon [50] also provide a powerful tool to exploit pulsed dual-comb optical sources for the on-chip generation and control of the multiple graphene plasmons generated by DFG, further boosting the practicality of this scheme on versatile device platforms.

>> I believe that the current manuscript is not suitable for Nature Nanotechnology, but for a sister journal.

Following the suggestion of the Referee, we have submitted a revised version of the manuscript to Nature Communications.

Reviewer #2 (Comments for the Author):

>> The manuscript by Li. et. al presents a demonstration of nonlinear excitation of graphene plasmons with counter-propagating ultrafast pulses through a difference-frequency-generation (DFG) process. Because of the electrical tunability of graphene plasmonic properties, the DFG process is also controllable. Through this controllability, the authors demonstrate the possibility to implement some logic operations.

We thank the Referee for the nice and concise summary of our work and for recognizing that we “demonstrate the possibility to implement some logic operations” with our device.

>> The overall fundamental concept is similar to a previous paper from the corresponding author (Nature Photonics volume 12, pages 22–28 2018), which reduces the novelty of current manuscript. The logic operation demonstration is quite proof-of-concept, and it is doubtful whether this can be a viable route for future logic integrated photonic devices for broader audience. The whole manuscript was written in a concise way and easy for readers to follow.

We agree with the Referee that the basic principle of DFG of graphene plasmons has been recently reported by us [Yao, *et al. Nat. Photonics* 12, 22, 2018]. However, as the Referee points out, our manuscript contains different aspect of novelty with respect to the simple DFG of graphene plasmons, including in particular the demonstration of multiple plasmons’ co-control for logic operations. We agree that this demonstration is proof-of-concept, however we still believe that the idea is worth publication and future works can focus on the engineering and optimization of our device for real-world applications.

>> Technically, I find there are multiple pieces of critical information/evidence missing to support claims made by the authors, from all fronts of electronics, optoelectronics, and nonlinear optics. The details are shown below.

We thank the reviewer for his/her constructive comments. In the following we carefully address point-by-point all his/her concerns.

>> Thus, in summary, I would not recommend the publication of this manuscript on Nature Nanotechnology. It could be more suitable for journals like Nature Communications.

Following the suggestion, we submit a revised version of the manuscript to Nature Communications.

Electronics:

1. How do authors achieve the tuning of Fermi level? In the diagram in Figure 1a and the photo shown in Fig. S7, two gold electrodes were deposited to seemingly behave as drain and source electrodes and it is a two electrical-probe experiment. I don’t see how Au-graphene-Au can behave as a transistor.

We thank the Referee for the question, that gives us the opportunity to address this important point. First, we would like to refer to one of our recent publications [Nano Letters 20, 6473 2020], where a similar geometry (graphene on a D-shaped fiber) was used. There, we explain in details how the Fermi level can be tuned in a bipolar junction (see e.g. figure 1d and corresponding text). This type of device, namely a graphene bipolar device, has been demonstrated and discussed in detail for instance in ref. [Nature

Nanotech. 5, 487 2010]. In brief, by tuning the gate voltage in a two-terminal device, the current flowing in the graphene channel does not increase linearly (as one would expect for a metal), but instead it shows a kink corresponding to the transition from n-transport to p-transport across the Dirac point (see also Phys. Reports 606, 1, 2016; Nature Nanotech. 12, 1, 2008). This is shown schematically in Fig. R1.

In particular, for $V_{sd} < V_{sd-kink}$, the current is dominated by holes (region I). For $V_{sd} = V_{sd-kink}$, the vanishing carrier density produces a ‘pinch-off’ region at the drain (region II) that renders the current in the channel less sensitive to V_{sd} and thus results in the pronounced kink seen in the I–V characteristic (figure R1 and S6). For $V_{sd} > V_{sd-kink}$, the minimum density point resides in the channel, producing a pinch-off region that moves from source to drain with increasing source-drain voltage (region III). In this bias range the carriers in the channel on the source side of the minimum density point are holes, and those on the drain side are electrons. The voltage drop across the ‘hole’ portion of the channel remains fixed at $V_{sd-kink}$, while the voltage drop across the ‘electron’ portion increases as $V_{sd} - V_{sd-kink}$. In this ambipolar regime, the pinch-off point becomes a place of recombination for holes flowing from the source and electrons flowing from the drain.

In the revised version of the manuscript, we have added the I-V curve of our device (new Fig. S6a) and the corresponding paragraph to further clarify this point. In addition, we have added the following sentence to the main text:

“Such behaviour is typical of a graphene bipolar junction [38,42], as discussed in detail in Supplementary Section S2.”

2. If carrier injection/depletion happens at Au-graphene-Au contacts, it seems to suggest a junction forms at the interface and current-voltage curves be nonlinear. Thus, it is unclear to me how authors obtain the graphene resistance and can tune the Fermi level simultaneously in a two-terminal device. The authors should provide current-voltage curves to show how resistance shown in Figure 1d is defined and obtained.

In the revised version of the manuscript, we provide the I-V curve of the device and its detailed description, as we explained in the reply to point 1. In addition, we would like to stress that the nonlinearity of the device that we observe in the I-V curve does not originate from the depletion junction between the metal and graphene but instead from the change in the transport properties (from holes to electrons) in the graphene channel, as detailed in the answer to question 1. The channel length of our device is $> 200 \mu\text{m}$, meaning that the channel resistance is higher than the contact resistance. So we can directly extract the resistance of graphene from the derivative of the I-V curve. We provide the related electrical analysis in the SI:

“We now discuss the electrical properties of the GDF device (Fig. S6). The possibility of tuning the Fermi level in a two-terminal device is a unique feature of graphene: its ambipolar behaviour results in a “kink” in the I-V curve [16]. Due to the external voltage (electron injection), the p-doped graphene on the fiber (SiO_2 substrate) becomes first neutral and subsequently n-doped. Thus, the “kink” corresponds to the transition from n-transport to p-transport across the Dirac point, suggesting that the resistance (or the carrier density) can be tuned by changing the source-drain voltage [17,18]. The blue curve in Fig. S6a plots the I-V curve of our device. This is clearly nonlinear: when the driving voltage V_{sd} is negative ($V_{sd} \ll V_{kink}$), the current is carried by holes throughout the length of the channel (Region I). When $V_{sd} \approx V_{kink}$, the vanishing carrier density produces a ‘pinch-off’ region at the drain (region II) that renders the current in the channel relatively insensitive to V_{sd} and results in the pronounced kink seen in the I–V characteristic. Here the graphene is close to the Dirac point. For $V_{sd} \gg V_{kink}$, the minimal density point resides in the channel, producing a pinch-off region that moves from source to drain with increasing voltage (region III). In this bias range the carriers in the channel on the source side of the minimum density point are holes, and those on the drain side are electrons. In this ambipolar regime, the pinch-off point becomes a place of recombination for holes flowing from the source and electrons flowing from the drain. In the bottom panel of Fig. S6a, we show the zoomed-in curve when V_{sd} is in a small range -0.1 V to 0.1 V . Here, the linear response clarifies that the kink is not induced by the graphene-Au based Schottky effect [19]. By using the expression $\sigma = dI_{sd}/dV_{sd}$, in Fig. S6a (red curve) we plot also the conductivity (σ) of our device.

The Fermi level of graphene can be written as $|E_F| \approx \hbar |v_F| (\pi N)^{-1/2}$, where N is the carrier density, \hbar is the reduced Planck’s constant and v_F is the Fermi velocity. Thus, one can calculate the E_F of a device as a function of V_{sd} , which changes the carrier concentration N based on the above discussion (reply to

point 1). In our device, we use $N = \epsilon_0 \epsilon_g / V - V_{Dirac} / ed$ to calculate the carrier density. Here ϵ_0 and ϵ_g are the permittivity in vacuum and in graphene respectively, V_{Dirac} represents the source-drain voltage needed to keep the graphene at the Dirac point (i.e, the kink in the I-V curve), e is the electron charge and d is the graphene thickness [20]. Therefore, the mobility of graphene can be determined using the Drude model $\sigma = eN\mu$, where σ is the conductivity, μ is the mobility, e is the elementary charge and N is the carrier density. The calculated mobility is shown in Fig. S6b and we obtain $\approx 1800 \text{ cm}^2 \text{V}^{-1} \text{ s}^{-1}$, a typical value for CVD samples [21].

Fig. S6 | Electrical measurements and characterization. a, I-V curve and conductivity of the GDF device, the bottom panel is a zoom of the source-drain current. b, Calculated mobility of the GDF.

3. Furthermore, I am not sure how the authors obtain the curve of Fermi level in Figure 1d. What equations do authors use to obtain these Fermi levels?

The Fermi level of graphene can be written as $|E_F| \approx \hbar |v_F| / (\pi N)^{-1/2}$, where N is the carrier density, \hbar is the reduced Planck's constant and v_F is the Fermi velocity. Thus, one can calculate the E_F of a device as a function of V_{ds} , which changes the carrier concentration N based on the above discussion (reply to point 1). In our device, we use $N = \epsilon_0 \epsilon_g / V - V_{Dirac} / ed$ to calculate the carrier density. Here ϵ_0 and ϵ_g are the permittivity in vacuum and in graphene respectively, V_{Dirac} represents the source-drain voltage needed to keep the graphene at the Dirac point (i.e, the kink in the I-V curve), e is the electron charge and d is the graphene thickness (see also Nature Nanotech 3, 210, 2008; Synth. Metals 244, 36, 2018; Nature Nanotech 3, 654, 2008 for further discussion). We have now added this explanation to the SI, in relation to Fig. S6.

It is surprising to me that a CVD-grown graphene transferred on substrates can achieve Fermi level = 0 eV (at Dirac point) with such a simple two-terminal electrical device.

We agree with the Referee that it is experimentally impossible to achieve $E_F = 0$ eV due to the presence of puddles and inhomogeneities, as discussed for instance in ref. [Pogna et al., ACS Nano 2022, <https://doi.org/10.1021/acsnano.1c04937>]. However, for simplicity we define $E_F = 0$ eV the point where we have the minimum conductivity, although strictly speaking this does not correspond to a perfectly undoped system.

We have now added this reference to the manuscript and we have also added the sentence:

“We notice that it is experimentally impossible to achieve $E_F = 0$ eV due to the presence of puddles and inhomogeneities [43]. However, for simplicity we define $E_F = 0$ eV as the point where we measure the minimum conductivity in the GDF.”

Moreover, as shown in Fig. S5a, the monolayer property of graphene is not that good (2D/G ratio is not that high) and the graphene is quite defective (D peak is large). The existence of grain boundaries, defects, and multiple layers in graphene make the tuning of Fermi level to the Dirac point challenging.

As mentioned above, we agree that the presence of defects and inhomogeneities contribute to larger variations in the doping (puddles). However, this does not change the main message of our manuscript, i.e. the tunability of the nonlinear co-generation of graphene plasmons. In other words, for our application it is not necessary to achieve exactly $E_F = 0$ eV. In the revised supplementary information,

we add more measured Raman spectra that further substantiate the quality of our CVD graphene sample.

“Besides the I-V measurement, we characterize the $|E_F|$ of our graphene device for different values of the V_{sd} voltage by means of in-situ Raman spectroscopy (Renishaw InVia: 514 nm laser excitation, on-sample power <1 mW, integration time 1 s). Fig. S7a plots three typical Raman spectra of our GDF device (no external voltage, measured at 3 different locations), the ratio of <1/2 between the G and 2D peaks (in Lorentzian shape) reveal the single layer nature of our graphene sample [22]. The half-width of the G peak at $\approx 1586 \text{ cm}^{-1}$ is $\approx 26 \text{ cm}^{-1}$, the 2D peak is at $\approx 2681 \text{ cm}^{-1}$ with FWHM $\approx 40 \text{ cm}^{-1}$. The wet-transfer technique based on-fiber deposition enables phonon coupling and induces defects at some positions. For example, we observe a clear D peak at 1340 cm^{-1} , a small D+D'' peak at 2462 cm^{-1} , and a minor D+D' peak at 2961 cm^{-1} , at the point shown in the top panel of Fig. S7a. These asymmetric phonon scattering peaks also contribute to the $\chi^{(2)}$ and to the plasmon-phonon coupling. But generally, D peak is smaller than $1.2 \times 10^3 \text{ a.u.}$, suggesting the graphene quality is acceptable [23].”

Fig. S7 | In-situ Raman spectroscopy measurements. a, Raman spectra of the GDF. The positions of the D, G, D', D+D', D+D'' and 2D peaks are marked on the graphs. b, Measured Raman spectra at different V_{sd} for the G peak (top) and the 2D peak (bottom). Here the V_{sd} is tuned from 0 V to 0.5 V and 1 V, corresponding to $|E_F|$ tuning from < 0.1 eV to $\approx 0.4 \text{ eV}$.

4. How do authors rule out any heating effect from graphene, which can behave as heaters (e.g., in this paper Optica 3, 159-166 2016)? Two-probe experiment is more likely to be generating heat inside the graphene. Also, the heating effect (temperature change in graphene) can also induce Raman spectra shift in Fig. S5b.

We thank the Referee for the question. We discuss this point in detail in the answer to question 11, which also refers to thermal effects. In brief, we show that the thermal effects induced by the source-drain voltage are negligible, because the driving voltage is relatively small ($\pm 1V$). Please see the reply to question 11 for more details.

5. What's the carrier mobility of graphene? The plasmonic wave optoelectronic properties of graphene are strongly dependent on the graphene mobility. This information is important to understand the loss mechanism inside devices.

We calculated the mobility of our graphene device from $\sigma = eN\mu$ (Nature Nanotech. 3, 210, 2008), where σ is the conductivity, μ is the mobility, e is the elementary charge, N is the carrier density. The calculated mobility is now shown in Fig. S6b, and we obtain $\approx 1800 \text{ cm}^2\text{V}^{-1} \text{ s}^{-1}$, a typical value for CVD graphene [see e.g., Sci. Adv. 1, 6, 2015; DOI: 10.1126/sciadv.1500222]. Please see the new Fig. S6.

Optoelectronics and nonlinear optics:

6. In the simulation shown in Fig. 1b, what component of electric field is plot? What is the sign and scale bar in Fig. 1b?

Fig.1b shows the $|E_y|$ component of the electric field. The simulated field intensity is normalized, and the spatial scale bar is $6 \mu\text{m}$. We have now added these details to the revised version of the manuscript.

7. It is stated in the manuscript “the graphene-fiber heterostructure has the loss $<0.5 \text{ dB}$ ”. Is it for telecommunication wavelength? It is clear that in the range of achievable Fermi level, from 0 eV to 0.4 eV, the loss in graphene is also strongly modulated. Particularly, at 0 eV Fermi level, the strong interband loss exists in graphene for telecommunication wavelength.

Yes, this value of the loss is measured at telecommunication wavelength and the 0.5 dB is the linear loss without considering interband absorption. We agree that the Fermi level tuning will change the optical transmission, as already shown in Fig. S5.

We have now revised the main text with the following paragraph:

“the total linear losses of the graphene-fiber heterostructure are $< 0.5 \text{ dB}$ at 1560 nm when the $|E_F|$ of graphene is $> 0.4 \text{ eV}$. Clearly, by tuning the graphene Fermi level, the transmission loss of the device will change. Further details on the nanofabrication and characterization of the GDF are provided in Supplementary Section S2.”

And in the SI:

“In Fig. S5a, we plot the measured optical transmission loss of our GDF device in the range 1500 nm to 1600 nm, obtained using a continuous-wave tunable laser (Santec TSL-710). In this measurement, we fix the average launched-in optical power to 1 mW (far below the Pauli blocking threshold to avoid saturable absorption) and vary the driving voltage. In the spectrum, the higher loss at lower frequencies (longer wavelengths) is due to the stronger evanescent scattering. Besides, when changing the driving voltage on the GDF, light transmission can be considerably tuned. This is determined by the electrically tunable nonlinear absorption. Such a phenomenon has been widely used in graphene based optoelectronic modulators [15]. In Fig. S5b, we plot the correlation between the driving voltage and the measured transmission for a fixed laser wavelength of 1560 nm. The transmission of our GDF decreases first from 26.7% (0 V, $E_F \approx 0.1 \text{ eV}$) to 23.2% (0.25 V, $E_F \approx 0 \text{ eV}$), and finally increases back to 88.1% ($> 1.25 \text{ V}$, $E_F > 0.4 \text{ eV}$). This process is due to the interband photoelectron transition. At least 10% ($\approx 0.5 \text{ dB}$) of the losses is induced by the linear absorption and scattering of the D shaped fibre.”

Fig. S5 | Optical transmission measurement. a, Measured transmission spectrum of the GDF in the range 1500 nm to 1600 nm. **b,** Electrically tunable transmission in the GDF at 1560 nm.

8. It is not clear to me what experimental evidence supports the authors’ claim that the graphene plasmonic excitation is involved in the DFG process. As described in the manuscript by the authors, the DFG process occurs when the dispersion of graphene plasmonic modes match phase-matching conditions. It sounds to me that such dispersions can be obtained experimentally by controlling the pump and probe, which should be provided to support authors’ claim (in addition to current simple RPA-based calculations).

We thank the Referee for the interesting question. Indeed, by tuning the pump and probe frequencies one can control the DFG process, which is defined by both the plasmonic dispersion and the phase matching. In Fig. 2, we experimentally show the configuration when the plasmonic dispersion is changed (this is a key point to realize optoelectronic logic operations). As shown in Fig. 1e, by changing the pump wavelength, the plasmonic difference frequency is also tunable. This has been already experimentally verified in our previous study (Nat. Photonics 12, 22, 2018). In this work, since

we use two synchronized laser frequency combs as the pump-probe pair, it is hard to tune their central wavelength on a large range. Nevertheless, we have now added more results in the supplementary section 3, where we change the central wavelength of the pump laser comb (see also revised figure S9).

“The phase matching for DFG of graphene plasmons has been already discussed in the past [10,24,25]. In order to further verify the phase matching conditions in our experiment using two pulsed light sources, we tune the central wavelength of the pump laser (λ_p) at 1530 nm, 1560 nm and 1590 nm (Fig. S9a) and measure the plasmon DFG for different values of the graphene E_F . Fig. S9b to Fig. S9d show the plasmonic oscillation peak of the lowest dispersion branch, when changing the λ_p from 1530 nm to 1590 nm and for $E_F = 0.1$ eV, 0.2 eV and 0.3 eV. It is clear that the f_{sp} shifts down when tuning the E_F . In particular, for increasing λ_p from 1530 nm to 1590 nm, when $E_F = 0.1$ eV, the f_{sp} decreases from 7.54 THz to 7.4 THz, when $E_F = 0.2$ eV, the f_{sp} decreases from 11.22 THz to 11.02 THz and when $E_F = 0.3$ eV, the f_{sp} decreases from 13.08 THz to 12.82 THz. In this figure, we also show the phase matching condition (bottom panels). Here the red curve plots the graphene dispersion in the GDF and the blue curves plot the calculated pump-probe dispersions. The dots show the measured phase-matched points, in good agreement with our model.”

Fig. S9 | Tuning of the pump wavelength and measurement of the dispersion change. a, Tuning of the pump laser combs, with central wavelengths 1530 nm, 1560 nm and 1590 nm. b to d, Measured plasmonic DFG. Panels from left to right are for $E_F = 0.1$ eV, 0.2 eV and 0.3 eV. In each sub-figure, the top panel shows the reproduced plasmonic oscillations, while the bottom panel shows the phase matching space.

Is it possible the DFG process occurs because of the existence of metal electrodes?

The co-generation of plasmons by DFG can’t occur at the metal contacts for two main reasons. First, the gold electrodes in our device are not in contact with the fiber’s core, hence there is no optical overlap that allows for DFG. Second, the plasmonic dispersion of gold is very different from that of graphene. For instance, at $k = 10^7 \text{ m}^{-1}$, the typical plasmon frequency of gold is > 300 THz, which is orders of magnitude higher than the f_{sp} of graphene. In order to further confirm that the metal electrodes do not contribute to the plasmon DFG, we repeated the same experiments on a D shaped fiber (including gold contacts) without graphene. As expected, we did not observe any plasmon generation in this case (see Fig. S11a). We have now added a section in the SI to discuss this point in detail. In addition, we have added the following text to the manuscript:

“For a D shaped fiber without graphene, we did not observe any plasmon generation nor DFG. This indicates that the process originates from graphene instead of the gold contacts (see Supplementary Section S3).”

In the SI:

“We also verify that the plasmons are generated in graphene rather than at the gold electrodes. The

DFG process can't occur at the gold contacts for two main reasons. First, the gold electrodes in our device are not in contact with the fiber's core, hence there is no optical overlap that allows for DFG. Second, the plasmonic dispersion of gold is very different from that of graphene. For instance, at $k = 10^7 \text{ m}^{-1}$, the typical plasmon frequency of gold is $> 300 \text{ THz}$, which is orders of magnitude higher than the f_{sp} of graphene. Fig. S11a shows the ΔI_{DFG} in a device without graphene (only gold contacts): here there is no evidence of plasmon generation."

Fig. S11 | a, DFG in an Au covered D shaped fiber. Top: Optical microscope image of the device, Bottom: ΔI_{DFG} , measured in the optical band 1600 nm to 2000 nm.

9. Following up the previous question, I am not sure how losses affect the excitation of tens of THz plasmonic wave in graphene. The carrier mobility of CVD graphene is not high and the loss from free-carrier absorption can be large at finite Fermi level. Both factors can lead to very lossy plasmonic wave in the range of tens of THz. For such lossy plasmons, I would expect their effect should be a minor effect.

We agree that losses play an important role in the plasmonic generation, as suggested also by the RPA simulations. However, our measurements clearly demonstrate that for the losses of our device (which include the mobility of our CVD graphene) it is possible to generate plasmons in graphene by DFG.

In addition, what value of L_{sp} is used for the DFG signal calculation? L_{sp} should be also a function of Fermi level.

It is true that L_{sp} is a function of the Fermi level (see also Nat. Photonics 12, 22 2018). Indeed, Fig. 2e already plots the ΔI_{DFG} (corresponding to the plasmonic intensity) in the GDF at different values of E_F . We now discuss this point more in details in the SI. In addition, we have now included the following paragraph in the main text:

“as expected for DFG considering $I_{sp} = \Delta I_{DFG}(f_{sp}/f_{probe}) = [\chi^{(2)}]^2 I_{pump} I_{probe} / L_{sp}^2$, where L_{sp} is the E_F dependent transmission loss of the plasmon (see Ref. [33] and Supplementary Section S1).”

and in the SI:

“Beside the nonlinear gain, we also consider the plasmon coupling based loss $L_{SP}(f,k)$ along the GDF, considering the phonon coupling. The $L_{SP}(f,k)$ is determined by both the carrier mobility and the Fermi level of graphene [12].

$$L_{SP}(k, f) = -\text{Im} \left\{ 1 - \frac{e^2}{2k\epsilon_1} \psi - \sum_j f_{ph,j} \psi \right\} \quad (\text{S9})$$

$$\psi = -\frac{g_s}{4\pi^2} \sum \int \frac{f_d(\epsilon_s) - f_d(\epsilon_{sk})}{2\pi f \hbar + \frac{\hbar}{\tau} + \epsilon_s - \epsilon_{sk}} dk F(s, k) \quad (\text{S10})$$

Here $f_{ph,j}$ are the phonon resonances, $g_s = 4$, $f_d(\epsilon)$ is the Fermi-Dirac distribution, $\epsilon_s = sv_F k$, $\epsilon_{sk} = sv_F k$, $s = \pm 1$, $F(s,k)$ is the band overlap function of the Dirac spectrum, which equals 1 for the waveguide geometry.”

10. The time-correlation measurements suggest that this is a coherent process. In my opinion, the claim that this measurement indicates the graphene device is promising for high-speed devices and logic operations is a bit over-selling. Such nonlinear processes (e.g., DFG) in any materials can show similar behavior in time-correlation measurements. I would suggest the authors tune down this claim.

We thank the Referee for the suggestion. In the revised version of the manuscript we have removed the claim related to high-speed logic operations in relation to the time-correlation measurements.

11. Also, how do the authors rule out any thermal effect in the DFG processes, especially when applying currents?

We thank the Referee for the question. The thermal effects induced by the source-drain voltage are negligible, as we clarify in the following, because the driving voltage is relatively small ($\pm 1V$). Considering the Joule heating $P = IV$, the typical mass of our GDF device ($3.3 \mu\text{g}$) and the typical heat capacity of the fiber ($800 \text{ J kg}^{-1} \text{ K}^{-1}$) [An, N. et al. *Nano Letter* 20, 6473, 2020], we can estimate the temperature increment as a function of the source-drain voltage. This is now plotted in the new Fig. S11b. Even for $V_{sd} = 1V$ (current in the channel 0.9 mA), the temperature increment is $< 260 \text{ mK}$. To show this, we placed the device in a controlled environment by placing a TEC in a vacuum chamber. Here, we can monitor the thermal image (using a thermal camera, FLIR) of the device, as we change V_{sd} . As expected, we observe that the temperature in the channel does not increase remarkably by tuning the source-drain voltage (see Fig. S11c).

We have now added the following paragraph to the SI:

“Another important aspect to consider is the thermal effect. The thermal effects induced by the source-drain voltage are negligible because the driving voltage is relatively small ($\pm 1V$). Considering the Joule heating $P = IV$, the typical mass of our GDF device ($3.3 \mu\text{g}$) and the typical heat capacity of the fiber ($800 \text{ J kg}^{-1} \text{ K}^{-1}$) [An, N. et al. *Nano Letter* 20, 6473, 2020], we can estimate the temperature increment as a function of the source-drain voltage. This is now plotted in the new Fig. S11b. Even for $V_{sd} = 1V$ (current in the channel 0.9 mA), the temperature increment is $< 260 \text{ mK}$. To show this, we placed the device in a controlled environment by placing a TEC in a vacuum chamber. Here, we can monitor the thermal image (using a thermal camera, FLIR) of the device, as we change V_{sd} . As expected, we observe that the temperature in the channel does not increase remarkably by tuning the source-drain voltage (see Fig. S11c).”

Fig. S11 | The role of gold contacts and thermal effects. **a**, DFG in an Au covered D shaped fiber. Top: Optical microscope image of the device, Bottom: ΔI_{DFG} , measured in the optical band 1600 nm to 2000 nm. **b**, Calculated Joule heating effect in the device. **c**, Measured thermal images and temperature curve. The error bar represents the measurement uncertainty of the FLIR ($\pm 10 \text{ mK}$).

Logic device/operations:

12. Overall, I appreciate the authors’ demonstration of logic operations. However, in my opinion, claiming this method can be used for future logic integrated photonic devices could be over-selling. Compared to the micro-ring based all-optical logic circuits, the demonstrated device, especially in general for graphene-based devices, the large-scale fabrication and manufacturing can be challenging.

Also, the use of strong ultrafast pulses is not favorable for chip-scale devices. I would suggest the authors tune down this claim and consider revising the title for future submission.

We thank the Referee for the positive comment and for acknowledging the novelty of our demonstration of logic operations. We agree that the current device on a D shaped fiber is hardly scalable, however we are convinced that the same principle (logic operations with DFG of plasmons) can be applied to other types of integrated photonic platforms. In this respect, we believe that our paper will generate a strong interest in the scientific community. In addition, we would like to stress that the state-of-the-art for heterogeneously integrated microcombs on silicon (Science 373, 99–103 2021) now offer a powerful way to incorporate pulsed frequency combs on-chip, thus making the use of ultrafast pulses favourable also for chip-scale devices. We now discuss this point more in details in the revised version of the manuscript:

“In future, the plasmonic co-generation and control scheme demonstrated here could be easily extended to on-chip integrated graphene optoelectronic devices suitable for large-scale fabrication, such as silicon nitride waveguides [33] and microrings [49]. Recent advances in the field of heterogeneously integrated microcombs on silicon [50] also provide a powerful tool to exploit pulsed dual-comb optical sources for the on-chip generation and control of the multiple graphene plasmons generated by DFG, further boosting the practicality of this scheme on versatile device platforms.”

The Referee acknowledges that we are able to perform logic operations, but he/she doubts that these are suitable for devices/gates. We thus follow his/her suggestion and modify the word “gate” with the word “operations” in the title. The new title now reads: “Nonlinear co-generation of graphene plasmons for optoelectronic logic operations”.

REVIEWERS' COMMENTS

Reviewer #2 (Remarks to the Author):

I would like to thank the authors for carefully addressing my comments in detail. Most of my comments have been fully addressed. I have two follow-up comments:

1. Question 6: I still don't see any labels on what electrical field is displayed in Fig. 1b or its caption. Please double-check.
2. Question 9: Can authors provide a few concrete numbers of L_{sp} (instead of equations) extracted or used in the demonstrated experiments?

After addressing these two minor comments, I would recommend the publication on Nature Communications.

Response to referees (NCOMMS-22-09613-T)

Reviewer #2 (Remarks to the Author):

I would like to thank the authors for carefully addressing my comments in detail. Most of my comments have been fully addressed. I have two follow-up comments.

After addressing these two minor comments, I would recommend the publication on Nature Communications.

Response: We sincerely thank the Referee for her/his comments and suggestions, which have greatly helped us to improve the quality of our manuscript.

1. Question 6: I still don't see any labels on what electrical field is displayed in Fig. 1b or its caption. Please double-check.

Response: We have added a color bar showing the electrical field intensity in Fig. 2b, and we also added this information in the caption.

Fig. 1. b, Simulated electrical field distributions of an optical mode propagating in the fiber core and a plasmonic mode propagating along the graphene-fiber interface (transverse magnetic polarization, E_y). SP mode: surface-plasmon mode. Scale bar: 6 μm . Color bar: normalized E_y .

2. Question 9: Can authors provide a few concrete numbers of L_{sp} (instead of equations) extracted or used in the demonstrated experiments?

Response: We thank the Referee for the suggestion. As discussed in the supplementary equations, the dimensionless parameter L_{sp} is related to both $|E_F|$ and energy-momentum matching. We now address this point more clearly.

(1) In the maintext, we added the following sentence:

Specifically, when the phase matching condition is satisfied, for $|E_F| = 0.1$ eV, 0.2 eV and 0.3 eV, $L_{sp} \approx 1.67$, 1.18 and 1.04. For instance, in experiments at $|E_F| = 0.3$ eV, $\chi^{(2)}$ reaches 10^{-2} m/V, and $[\chi^{(2)}]^2/L_{sp}^2$ is on sub 10^{-4} W^{-1} level, in agreement with theoretical values [30].

(2) In supplementary section 1, we added the quantitative correlation between L_{sp} and $|E_F|$.

In the simplified case without considering the phonon effects, we plot the phase matched ' $|E_F|$ - L_{sp} ' correlation for the fixed pumping wavelength 1560 nm, as **Fig. S3b** shows.

Fig. S3. b, Under phase matching condition, L_{sp} decreases for increasing $|E_F|$.